# Vaccination Status and Attitude among Measles Cluster Cases in Austria, 2019

**DOI:** 10.3390/ijerph17249377

**Published:** 2020-12-15

**Authors:** Lukasz Henszel, Elisabeth E. Kanitz, Andrea Grisold, Heidemarie Holzmann, Stephan W. Aberle, Daniela Schmid

**Affiliations:** 1European Programme for Intervention Epidemiology Training (EPIET), European Centre for Disease Prevention and Control (ECDC), 169 73 Stockholm, Sweden; 2Austrian Agency for Health and Food Safety (AGES), 1090 Vienna, Austria; elisabeth.kanitz@ages.at (E.E.K.); daniela.schmid@ages.at (D.S.); 3Institute of Hygiene, Microbiology and Environmental Medicine, Medical University of Graz, 8010 Graz, Austria; andrea.grisold@medunigraz.at; 4Center for Virology, Medical University of Vienna, 1090 Vienna, Austria; heidemarie.holzmann@meduniwien.ac.at (H.H.); stephan.aberle@meduniwien.ac.at (S.W.A.)

**Keywords:** measles, infectious diseases, infectious disease transmission, vaccination, vaccination refusal, Austria

## Abstract

On 21 January 2019, public health authorities of two neighboring Austrian provinces reported an increase in measles cases. We investigated this occurrence to identify clusters of epidemiologically linked cases and the associated vaccination status in order to generate hypotheses on those factors explaining the size of the measles clusters. Probable cases were residents of the provinces of Styria or Salzburg with clinical presentation of measles after 1 January 2019 who were linked to a confirmed case using RNA virus detection. We collected data on age, rash onset, certificate-based vaccination status and reasons for being unvaccinated. Contact history was used to identify chains of transmission. By 11 March, we identified 47 cases, with 40 (85.1%) in unvaccinated patients. A cluster of 35 cases with a median age of seven years (IQR: 1–11) occurred between 9 January and 20 February in the province of Styria due to one transmission chain with four case generations. Of 31 vaccine-eligible cases, 25 (80.6%) were unvaccinated, of which 13 refused vaccination. Between 10 January and 1 March, we identified 12 cases as part of five unlinked clusters in the province of Salzburg. Each of these five clusters consisted of two generations: the primary case and the successive cases (median age: 22 years, IQR: 11–35). Eleven of 12 cases occurred in unvaccinated patients, with none of the 11 having a vaccination-refusing attitude. An extended measles cluster in a vaccination-refusing community, compared to five short-lived clusters concurrently occurring in the neighboring province, illustrates how vaccine refusal may hamper control of transmission.

## 1. Introduction

Measles is an acute, highly contagious viral disease with a high basic reproduction number (R_0_), often cited to be 12–18, that is capable of causing large epidemics [1]. Unimmunized children and susceptible adults exposed to the measles virus are at risk of severe disease and death due to complications such as pneumonitis and encephalitis [2]. Immunization is the only effective preventive measure against measles [3].

In 2012, the World Health Organization (WHO) set a target for measles to be eliminated by 2020 in all WHO regions [4]. At the end of 2018, only 35 countries in the WHO European region had obtained measles elimination status, fewer than in the previous years. In the period between 2018 and 2019, the steady increase in measles cases in the WHO European region reflected high measles activity in many countries, including Ukraine, Russian Federation, Serbia, Turkey, the Republic of North Macedonia, Romania, Italy, Bosnia and Herzegovina and Poland. Austria is a frequently chosen destination for immigration or short-term visits by citizens from the aforementioned nine countries, which poses the potential risk of measles importation [5,6]. The measles genotype B3 and/or D8 lineages were identified in eight of the above countries in 2018 [7] and in Austria in 2019.

Elimination of measles requires an immunity level of 95% in the population (i.e., herd immunity threshold). Immunity gaps allow (re-)introduction of the measles virus and local transmission [8]. One of the major reasons of low vaccine uptake among the European population is vaccine hesitancy, defined by WHO as a “delay in acceptance or a refusal of vaccines despite availability of vaccination services” [9]. The approaches in understanding vaccine hesitancy and acceptance differ. One is the 3C model, which focuses on influencing factors such as confidence in the vaccine or the vaccine provider, complacency towards the value of vaccination, and convenience (practical barriers) [10,11,12]. Another approach is the 5C scale, which assesses the psychological antecedents of vaccination: confidence, complacency, constraints (e.g., psychological barriers), calculation (related to extensive information searching) and collective responsibility [11]. Vaccine-hesitant individuals are a heterogeneous group including those who accept some vaccines, those who accept all vaccines but at a delayed schedule, and those who refuse all recommended vaccines [13]. Therefore, there is a need for targeting these different groups of vaccine hesitancy with specific strategies.

In Austria, the incidence of measles has decreased since introduction of the measles vaccine in the national immunization program in 1974. Between 2004 and 2007, Austria was considered a low- to moderate-incidence country according to the criteria of EUVAC.NET (<1/100,000 population/year) [14]. In 2008, a measles outbreak in the Austrian province of Salzburg, originating among the anthroposophic community, changed Austrian status to a high-incidence country [15]. In the following years, incidence ranged from 0.3 per 100,000 in 2016 to 3.6 per 100,000 in 2015. In 2018, Austria achieved elimination status, having demonstrated that endemic transmission of measles had been interrupted for 36 months [7]. In 2019, there were 151 measles cases reported, almost twice as high as the number of cases (n = 77) in 2018 [16].

On 21 January 2019, the public health authorities of two neighboring provinces (Styria and Salzburg) reported 10 confirmed measles cases, which had occurred since 9 January. This exceeded the expected number for the month January as compared to previous years. On 25 January, the Austrian Agency for Health and Food Safety (AGES) received the mandate from the provincial public health authorities of Styria and Salzburg to investigate this increase in measles cases. The first objective of the study was to describe clusters of measles cases by time, place and person, and if data were available to establish the chain of transmission. The second objective was to generate hypotheses on determinants for the magnitude of the measles clusters.

## 2. Materials and Methods

### 2.1. Cluster Case Definition

A confirmed cluster case was defined as a patient (1) who is a resident in the province of Styria or Salzburg, Austria, (2) with a generalized maculopapular rash onset between 1 January and 3 March 2019, accompanied by fever and cough, coryza, or conjunctivitis and (3) who fulfilled one of the criteria of laboratory-confirmed measles as described elsewhere [17]. A probable case was defined as a person with an epidemiological link to a confirmed case and fulfilling the clinical criteria for measles occurring between 1 January and 3 March 2019. Based on available data on contagious contact among cluster cases, we established the chain of transmission.

### 2.2. Case Finding and Data Collection

We identified cases associated with clusters by cross-checking against the measles cases recorded in the national surveillance database on measles, reported to the national electronic reporting system for all mandatorily notifiable diseases (*Epidemiologisches Meldesystem*), since 1 January 2019. We generated a case line list and intensified the case finding through interviewing known cluster cases with rash and fever among household members, in the same classroom, and in other community settings, and through monitoring contacts for clinical symptoms compatible with measles. We extracted cases information on age, gender, residence, date of rash onset, hospitalization, epidemiological link (defined as household membership, classroom-mate, workplace colleague, acquaintance), history of travel within three weeks prior to rash onset, and vaccination status. The vaccination status was categorized as ”receipt of two vaccine doses”; ”receipt of one vaccine dose”; ”no vaccination pre-exposure”, based on vaccination records; the “receipt of post-exposure vaccination”, further broken down into vaccination within or after 72 h following contagious exposure. We obtained information on the attitudes of cases or their legal guardians towards vaccination, including possible barriers to vaccination, through the telephone interview conducted by the local public health offices. We divided cases into three groups according to their different attitude towards vaccination:vaccine refusal—defined as an attitude of an individual who refuses all vaccines,vaccine criticism—defined as an attitude of an individual whose concern about vaccines may result in the refusal or the delay of some vaccines, andlack of knowledge on vaccination—defined as an attitude of an individual who does not have sufficient knowledge about preventive vaccination, including vaccination plans, which may result in failure of vaccine uptake in a scheduled time.

### 2.3. Laboratory Methods

Laboratory diagnosis of a measles virus infection was performed by serologic detection of specific IgM and IgG antibodies using commercially available ELISA tests. On serologically identified measles cases, measles RNA detection was also conducted using real-time PCR (LightMix^®^ Modular Measles Virus, TIB MOLBIOL GmbH, Berlin, Germany) in nasopharyngeal swabs and/or urine and/or serum samples, which were sent by local physicians, hospitals or local laboratories for confirmation and strain analysis to the National Reference Centre for Measles, Center for Virology, Medical University of Vienna. Genotyping was performed according to the measles and rubella WHO reference laboratory recommendations [18] using the Measles Nucleotide Surveillance (MeaNs) database tool for sequence analysis of a 450 nt amplicon coding for the nucleoprotein (N-450).

## 3. Results

### 3.1. Descriptive Findings

It was found that 47 cases (including 41 confirmed and 6 probable) of the total of 59 measles cases reported between 1 January and 11 March 2019 in Austria fulfilled the cluster case definition (Figure 1). Out of these 47 cases, 40 (85.1%) were unvaccinated. Out of the 43 vaccine-eligible cases, 36 were unvaccinated, four were vaccinated with two doses of MMR, and three were vaccinated with one dose of MMR. Out of 47 cases, 16 (34%) were healthcare associated, but were not healthcare workers. We identified clusters in two neighboring Austrian provinces: one in Styria (declared as cluster A) and five in Salzburg (declared as clusters B–F) (Figure 2), affecting communities with different attitudes towards vaccination. The remaining measles cases occurred as singletons (including imported cases) all over Austria.

#### 3.1.1. Cluster A

In Styria province, 35 identified cases belonged to cluster A, with one transmission chain with four case generations. The primary case, with rash onset on 9 January 2019, was an unvaccinated school pupil of 15 years of age with an anthroposophical family background and who consulted a pediatric clinic in Graz, Styria, during the contagious phase. In the same province, 34 successive cases occurred between 15 January and 20 February 2019 (Figure 1). Secondary (13 persons), tertiary and quaternary cases in the transmission chain occurred, but it was not possible to discriminate between third and fourth generation cases. The median age of the 35 cases was seven years (IQR: 1–11). Cluster A cases occurred most frequently in age groups 10–14 (n = 9), 1–4 (n = 8) and 5–9 (n = 8) (Figure 3).

Of the 31 vaccine-eligible cases in cluster A, 25 (80.6%) were unvaccinated, four (12.9%) were vaccinated with two doses of MMR, and two (6.5%) were vaccinated with one dose of MMR. Four cases were infants younger than nine months of age who therefore were not yet eligible for MMR vaccines. Thirteen (52%) out of 25 unvaccinated patients were vaccine refusers (Figure 4 and Figure 5). Twenty-six out of 35 cases had Austrian nationality. The remaining cases had: Belarusian (n = 1), Croatian (n = 2), Romanian (n = 1), Serbian (n = 1), Slovenian (n = 1), Hungarian (n = 1) and American (n = 1) nationalities. In one case, nationality was unknown. Post-exposure prophylaxis was administered in four cases.

#### 3.1.2. Clusters B–F

In Salzburg province, between 10 January and 1 March 2019, we identified 12 cases belonging to five further, unlinked clusters (clusters B–F) with chains of transmission, each of which subsided after two generations (5 primary, 7 secondary cases). The primary case of cluster D, with rash onset on 21 January 2019, aged 28 years, was an employee of the service sector. The primary cases of clusters B, C, E and F (6, 1, 48 and 27 years of age, respectively) had the date of rash onset on 10 and 11 January and 2 and 15 February 2019, respectively. No epidemiological link between the aforementioned clusters (B–F) has been identified. The median age of the 12 cases was 22 years (IQR: 11–35). As illustrated in Figure 3, the highest number of cases in clusters B–F occurred in age groups 1–4, 15–19, 20–24, 25–29 and 20–24, respectively (two cases in each age group). Of the 12 vaccine-eligible cases in Salzburg province, 11 (91.7%) were unvaccinated and one was vaccinated with one dose of MMR. None of the 11 unvaccinated patients had a vaccination-refusing attitude and five had a vaccine-critical attitude (Figure 4 and Figure 5). Ten out of 12 cluster cases had Austrian nationality, one was Bosnian and for one nationality was unknown. Post-exposure prophylaxis was administered in one case.

### 3.2. Laboratory Findings

For the viral detection, 41 collected specimens were positive. The viral RNA was extracted and the genotype variant D8-4683 (the MeV strain D8-Gir Somnath) was identified in 34 cases, including 28 cases belonging to cluster A and six cases to clusters B–D.

## 4. Discussion

During 2019, 151 measles cases occurred in Austria and the country faced multiple measles clusters. We report here on one cluster in Styria and five clusters in Salzburg province in the first three months of the year 2019 (Figure 6). These clusters occurred primarily among unvaccinated individuals and affected communities with different attitudes towards vaccination. In 2018, despite intensive public education campaigns, the coverage of the two-dose measles vaccine in two- to five-year-olds in Austria was estimated to be 82%, which makes the country vulnerable to community transmission following the introduction of measles [16]. In addition, the age distribution of measles cases indicates measles immunity gaps in adults [7]. We showed that despite implemented and ongoing control measures on a national level, (e.g., lowering the age of the first dose of MMR to nine months of age, recommended since 2015), there is a still a risk of measles infection due to remaining pockets of population subgroups with low immunization coverage. Overall, Austria is still among the European countries with the lowest measles vaccination rates, together with France, Malta, Greece and Romania, with coverage rates of the second dose of the measles vaccine below 95% in 2018 [7,16,19]. As a consequence, measles outbreaks are still likely to occur in Austria.

In a low-endemic country such as in Austria, outbreaks of measles have been repeatedly described in communities with low vaccination coverage due to philosophical or religious beliefs [20]. The reported cluster in Styria in 2019 originated from a case with an anthroposophical family background. Living in various European countries, anthroposophist communities have lower vaccination coverage and voice similar critical attitudes towards measles immunization [21]. Among the Austrian anthroposophic community, 94% of parents confess to the belief that a measles infection is important for a child’s development [20].

The occurrence of the circulating strains of genotype D8 in Austrian clusters in 2019 reflects the main reported genotypes in Europe. The primary case in Styria in the period preceding the occurrence of the disease stayed at a ski resort in Salzburg, which is frequently visited by tourists from Ukraine. The country was experiencing a large epidemic of measles in 2018, with D8 and B3 genotype variants identified and reported through Measles Nucleotide Surveillance (MeaNS), including D8-Cambridge, B3-Kabul, B3-Dublin and D8-Gir Somnath [7]. The last genotype variant was identified in the described Austrian clusters (a–d), including their primary cases.

According to the WHO, one of the ten most important health problems in many developed countries is vaccine hesitancy. Furthermore, the MMR vaccine is the vaccine that generates the most hesitation [22]. It is recognized that vaccine hesitancy contributed to the prolonged measles outbreaks that occurred in recent years in Europe [7,23] and in the US (defined as vaccine refusal). The resistance to immunization contributes significantly to the ongoing resurgence of transmission in Western Europe [24]. Based on the findings of the studies on the US-based measles outbreaks, vaccine refusal (as measured by population-level vaccine exemption rates) was associated with an increased risk for measles among people who refuse vaccines and among fully vaccinated individuals [25]. We found that ring vaccination was hardly accepted in Styria province, where an extended measles cluster occurred and 52% of unvaccinated cases were vaccine refusers. In contrast, in Salzburg province only short-lived transmission chains were identified, and none of the unvaccinated cases had a vaccination-refusing attitude.

Concerning our second objective, it is noteworthy that clusters among individuals with vaccine-hesitancy or lack of knowledge of vaccination were short-lived compared to the large cluster in a vaccine-refusing community in Styria. A vaccine-refusing attitude in a community may have an effect on the adherence to non-pharmaceutical measures for controlling virus spread, such as isolation, co-operation in the official contact tracing, quarantine of contacts, and even a misbelief that measles virus circulation benefits the health of an individual. However, the small number of clusters assessed is a limitation of our study.

To stop transmission chains and control measles outbreaks, it is important not only to notify unvaccinated people but also to implement innovative strategies to raise awareness among the population. The information communicated during the reported clusters in Austria may have influenced the public beliefs that the threat was severe, unvaccinated individuals were at risk and the measles vaccine was effective [26]; however, this was not sufficient to convince all vaccine-hesitant groups. Vaccine hesitancy can only be mitigated through a multidisciplinary and continuous strategy that must involve transparent communication of the underlying scientific evidence. This may not be sufficient to convince all vaccine-hesitant people, and the communicated evidence needs to be complemented with stories highlighting the risks of the disease and the social benefit for others, especially those who cannot be vaccinated [11,27].

In order to develop an effective strategy to improve confidence in vaccines in Austria, we need to understand the attitudes toward vaccination in hesitant subgroups and conduct surveys in the surrounding communities of clusters to examine their attitudes towards vaccination, the vaccine-preventable disease, and non-pharmaceutical interventions in general. Interventions must be adapted to the specific political, social, cultural and economic context of the population subgroups and must take into account that the number of factors influencing vaccine acceptance is dynamic and evolves over time [11,28,29]. Qualitative studies assessing determinants of vaccine acceptance might be more effective in the comprehensive understanding of how parents think and act about vaccination and might ensure that the intervention is tailored to the issue and the audience [28]. Qualitative studies may help promote vaccine acceptance by identifying the more favorable approaches to educate parents who are hesitant about vaccines [30,31] or by investigating the views of health workers on the barriers to and drivers of positive childhood vaccination practices [32].

## 5. Conclusions

An extended measles cluster in a vaccination-refusing community, compared to five short-lived clusters concurrently occurring in the neighboring province, illustrates how vaccine refusal may hamper the control of transmission. This hypothesis on vaccine attitude as one of the determinants for the magnitude of measles clusters has to be further tested.

Because declared vaccine refusers are generally not receptive to information campaigns, further qualitative studies are needed to be able to tailor interventions to address the specific concerns of vaccine-hesitant groups in a given context and time. We recommend information campaigns by the national health authority that are highly adapted to the Austrian vaccine-hesitant population groups as a way to increase knowledge and vaccine confidence; these campaigns can be supported by providing easy access to vaccines and the use of reminder systems. The national health authority should implement supplementary vaccinations to close those immunity gaps that are due to lack of knowledge and should embark on information campaigns to reduce the subgroups of vaccine refusers in the Austrian population.

## Figures and Tables

**Figure 1 ijerph-17-09377-f001:**
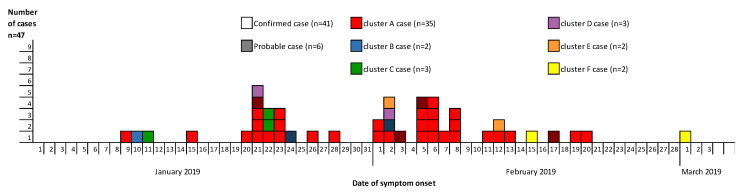
Measles (confirmed and probable) cases according to date of symptom onset, Austria, January–March 2019 (n = 47).

**Figure 2 ijerph-17-09377-f002:**
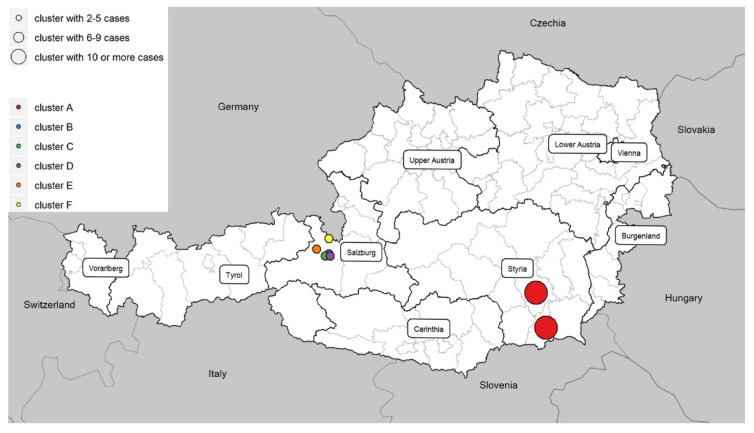
Measles cases according to day of symptom onset and cluster distribution, Austria, January–March 2019 (n = 47).

**Figure 3 ijerph-17-09377-f003:**
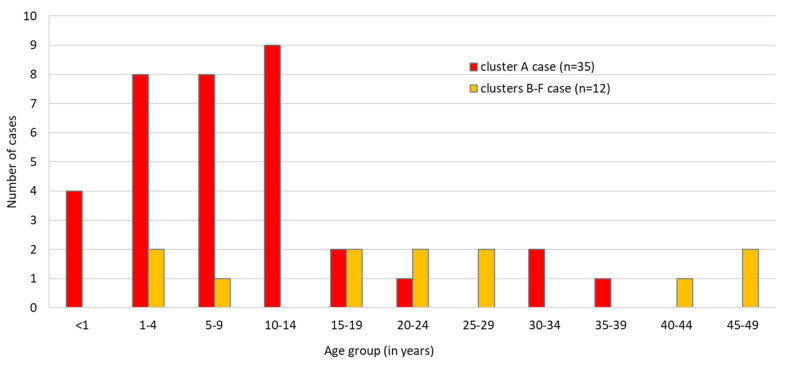
Measles cases according to age group and cluster, Austria, January–March 2019 (n = 47).

**Figure 4 ijerph-17-09377-f004:**
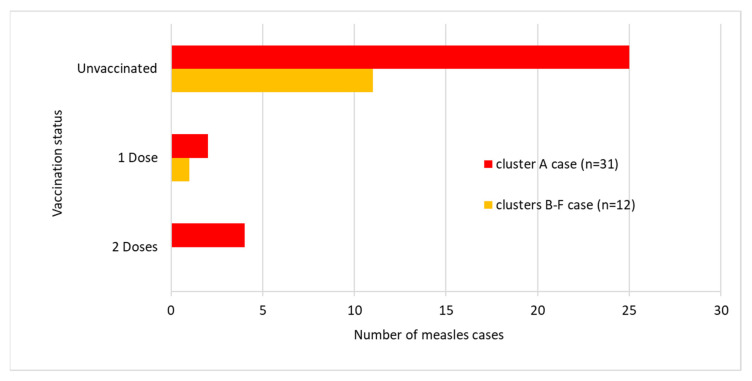
Measles vaccine-eligible cases according to vaccination status and cluster, Austria, January–March 2019 (n = 43).

**Figure 5 ijerph-17-09377-f005:**
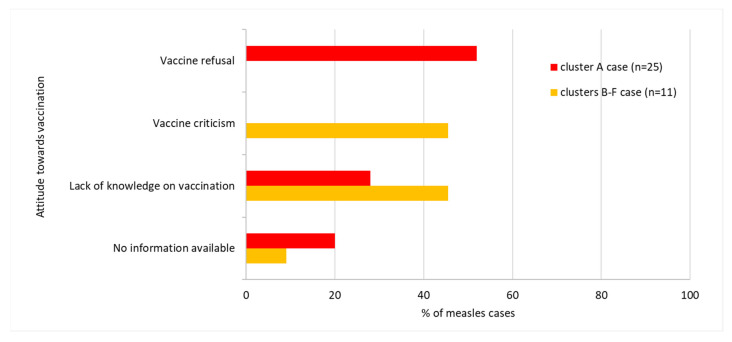
Measles cases according to attitude towards vaccinations and cluster, Austria, January–March 2019 (n = 36).

**Figure 6 ijerph-17-09377-f006:**
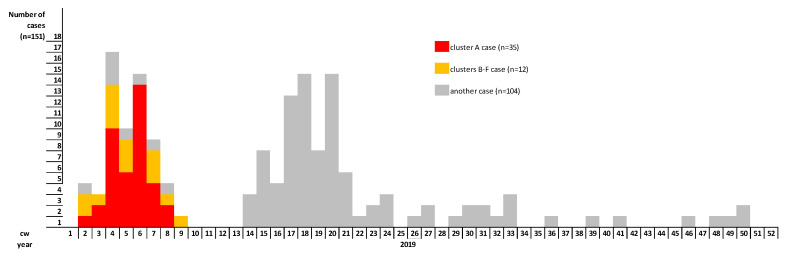
Measles (confirmed and probable) cases according to calendar week of symptom onset and cluster allocation, Austria, 2019 (n = 151).

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
