# Peer review of "Vaccination Status and Attitude among Measles Cluster Cases in Austria, 2019"

_ijerph, 2020, doi:10.3390/ijerph17249377_

Round 1

Reviewer 1 Report

The manuscript describes 47 measles cases that were identified in two provinces in Austria January to March 2019. The authors divide these cases into six separate outbreaks and compare age groups and vaccine attitudes between the largest outbreak (35 cases) in the province of Styria and the remaining five outbreaks in Salzburg province.

Measles viruses can be assigned to one of 24 genotypes based on the sequence of 450 nucleotides of the nucleoprotein gene. Sequence information was available for 34 cases. All shared the exact same sequence; they all belong to a virus lineage with global circulation in 2019.

The authors present no data to explain why the cases were divided into different outbreaks. A possible import source is described only for the first case in Styria, that source was a ski resort in Salzburg province, leading me to conclude that the cases in the two provinces were linked from the beginning. None of the other ‘outbreaks’ have import information, or epidemiological information justifying the assignment of the cases to independent lines of transmission. Cases in both provinces occurred simultaneously over a relatively short period of time. All sequenced cases belong to the same lineage. All this leads to the conclusion that there was one outbreak with cases in two provinces.

This leaves us with little useful information. The outbreak spread to two different communities: one (Styria) with anthroposophical beliefs who reject vaccines on principle and where mostly children were affected. The other community (Salzburg) included adults and children with other reasons for vaccine hesitancy which are not described. The authors write extensively about vaccine hesitancy, citing existing literature. However, no data or conclusions are presented that would make the different types of vaccine hesitancy in these two communities informative. The conclusion that the ‘outbreaks’ in Salzburg were more limited due to the less stringent vaccine hesitancy is not supported by data given that there is no proof these were separate outbreaks.

Author Response

Response to Reviewer 1 Comments

Thank you for your feedback. Please, see the answers in attachment.

Reviewer 2 Report

I think this is a well written paper, with adequate background, clear methodology, very clearly laid out findings, and conclusions that go with the findings. I recommend the paper for publication.

The manuscript is a descriptive study of an outbreak investigation. The introduction is very well written and describes the context of measles elimination in Europe, provides the incidence of measles in Austria in recent years.

The method utilised is standard epidemiological investigation of an outbreak, along with tracing of cases and an investigations into the beliefs and attitudes of the involved cases (or the parents of affected children) regarding vaccines. Clear definitions are provided of epidemiological terminologies used.

In the results section, the geographic areas, age breakdown, vaccination status and temporal trends of the distinct outbreaks have been described very well, including the viral genotypes involved. The charts and spot map are well labelled and clear enough to understand.

The authors discuss the implications of the outbreaks, and the reasons thereof in clear terms. they compare the genotypes detected in the outbreak to other documented outbreaks in Europe with the same genotypes. They have highlighted vaccine hesitancy as a reason for measles outbreaks. The issue is very pertinent and more and more an issue for the European Region in terms of attaining measles elimination status.

The references are pertinent and add to the understanding of the issues mentioned in the introduction and the discussion sections in the manuscript.

I did not find any methodological flaws in the study. The graphics are very clear. The results have been very clearly displayed, and the relevance of the results for the bigger agenda of measles elimination is discussed well. There are no issues with the language, as far as I could tell.

Author Response

Response to Reviewer 2 Comments

Thank you for your feedback. Please, see the answers in attachment.

Reviewer 3 Report

The article is generally clearly written and has great programmatic importance for the measles elimination program in Europe. It characterizes populations susceptibles to the measles virus, particularly due to the refusal of vaccination.
However, the purpose of the article is not clearly defined at the end of the introduction. Moreover, I strongly suggest that the last two paragraphs of the introduction (lines 70 to 82) be replaced by the exact description of the objective of the article.The cited lines can be moved into teh discussion chapter.

Otherwise the introduction contains enough information and references to understand the context.

Regarding materials and methods, I suggest that the current subchapters 2.2 and 2.3.  be combined into a single subchapter 2.2Data collection. I strongly suggest to clearly define the period chosen for data collection e.g. January 1 to March 31, 2019. Also it is not clear whether the information on attitudes towards immunization is included in a form, or whether patients and legat tutors were interviewed by the mission mandated by the MoH.
Also, in Chapter 2.4 Laboratory methods, it is not clear whether the laboratory first tested the sera for measles IgM before testing the samples for viral RNA. The WHO laboratory manual is not cited in the bibliography, so it is not clear whether the Reference Laboratory uses the standard methods described therein. If this is the case I suggest replacing the current reference 18 with the WHO manual.

The analysis of the epidemiological data collected is fairly well reported in the results chapter. However, since the objective of the study is not very well defined, I do not know whether it is a comprehensive article including the characterization of the population and the sequencing results, or this article is a descriptive study of the affected population only.
In the first case there is a lack of laboratory results such as: details on the number of samples received at the laboratory: how many nasopharyngeal swabs and the geographical origin, ? How many urine samples, ect... How many positive samples by sample type and geographical origin, ect... A descriptive table would be useful. The last column can contain the identified genotypes. One dendrogram per identified genotype should be added.
In the second case, it would be sufficient to add the genotypes identified in each district in Figure 2.
It is imperative to clearly define the purpose of the article.
Other suggestions
Figure 1: Hatching to identify the number of cases confirmed in the laboratory should be made more visible.
Figure 2: Add the names of neighbouring countries and the genotypes circulating in the two districts affected by the epidemic .
Figure 5; Add in the legend the total number of cases by outbreak A and B-F.
Lines 169-170 the sentence: "The median age of 35 cases was 7 years (IQR: 1-11). "it is not at the appropriate palce as the toatal number of cases for outbreak B-F is 12. Deleted or move it to outbreack A results description.

Regarding the discussion, here again it is difficult to assess it without a well-defined study objective:
If the article is purely a description of the cases, it is well done.
If the article is to include the molecular epidemiology of measles virus strains, then it is incomplete.
Same remark for the conclusion.

Author Response

Response to Reviewer 3 Comments

Thank you for your feedback. Please, see the answers in attachment.

Round 2

Reviewer 1 Report

An unvaccinated measles patients’ reasons for not getting vaccinated have no influence whatsoever on their ability to transmit virus. To say that one cluster of cases remains smaller than another based on the infected person’s attitude towards vaccines is looking at it backwards. The smaller clusters remained small because the virus found no new victims. It’s the people who remain uninfected and their attitudes towards vaccination who determine the size of an outbreak. If the authors wish to explain the size of clusters with attitudes towards vaccines, they need to examine the attitudes of the community and evaluate levels of under-vaccination in the affected communities. A community’s attitude towards vaccination will affect the level of under-vaccination before the outbreak and will affect acceptance of outbreak control measures (e.g. vaccination clinics) after the outbreak. What is the percentage of unvaccinated children and adults in the Styria anthroposophical community vs the Salzburg communities?  What measures were taken to control the outbreak and how did the communities react to them? Extrapolating from the attitudes of 12 measles cases in Salzburg province to the attitudes of the community they live in does not constitute a rigorous scientific study.

The authors have edited the manuscript to avoid the interpretation that these clusters were separate outbreaks but have not presented any new data to explain the size of the clusters. I still do not think this manuscript contributes significant new knowledge to the existing literature about measles outbreaks or vaccine hesitancy.

Author Response

Point 1: An unvaccinated measles patients’ reasons for not getting vaccinated have no influence whatsoever on their ability to transmit virus. To say that one cluster of cases remains smaller than another based on the infected person’s attitude towards vaccines is looking at it backwards. The smaller clusters remained small because the virus found no new victims. It’s the people who remain uninfected and their attitudes towards vaccination who determine the size of an outbreak. If the authors wish to explain the size of clusters with attitudes towards vaccines, they need to examine the attitudes of the community and evaluate levels of under-vaccination in the affected communities. A community’s attitude towards vaccination will affect the level of under-vaccination before the outbreak and will affect acceptance of outbreak control measures (e.g. vaccination clinics) after the outbreak.

I still do not think this manuscript contributes significant new knowledge to the existing literature about measles outbreaks or vaccine hesitancy.

Thank you for your feedback. Please, see the answers to your comments below. The changes in the manuscript are marked in yellow. Please note that we have changed the title of the manuscript. Please note that the relevant changes to the text of the manuscript are referred to in the abstract.

Response 1: We agree with the reviewer on the limitations mentioned, and therefore in this manuscript, we describe measles cases belonging to several clusters in terms of the vaccination status and different attitudes towards vaccination, in order to generate hypothesis which would need to be studied further with an appropriate study design, i.e. cohort study. We changed the title, which now reads as follows:

Vaccination status and attitude among measles cluster cases in Austria, 2019 (line 2-3).

Point 2: What is the percentage of unvaccinated children and adults in the Styria anthroposophical community vs the Salzburg communities? 

Response 2: In 2018, in general population, the coverage of 2-doses measles vaccine in two to five year old in Austria was estimated to be 82%. We do not have data on the anthroposophical community vaccination coverage from 2018.

A large outbreak in Salzburg in 2008 may have led to an elevated immunization thanks to mass vaccinations and increased natural immunization. Furthermore, in 2019 in the province Styria, general practitioners acting as influencers of strong vaccine refusing belief were demasked in the first half of 2019, after years of activism in the region.

Point 3: What measures were taken to control the outbreak and how did the communities react to them?

Response 3: A MMR post-exposure prophylaxis was offered free of charge to susceptible contacts of outbreak cases. A post-exposure prophylaxis was administered in four cluster A cases and 1 cluster B-F case. The supplementary MMR vaccination campaign to close immunity gaps due to lack of knowledge and information campaigns to reduce the subgroups of vaccine refusers in the Austrian population were recommended. The ring vaccination was hardly accepted in Styria province, where an extended measles cluster occurred and 52% out of unvaccinated cases were vaccine refusers. Please see the lines: 158, 177-178, 22-224, 267-268.

Point 4: Extrapolating from the attitudes of 12 measles cases in Salzburg province to the attitudes of the community they live in does not constitute a rigorous scientific study.

Response 4: We agree with the reviewer on this point, and therefore added a paragraph describing this limitation in the discussion section and also mentioned this notion in our conclusion. Based on the descriptive epidemiological study, we generated a hypothesis, which should be tested.

The new paragraph reads as follows:

Concerning our second objective, it is noteworthy that clusters among vaccine-hesitant individuals were short-lived compared to the large cluster in a vaccine-refusing community in Styria. A vaccine refusing attitude in a community may have an effect on the adherence to non-pharmaceutical control measures, such as isolation of cases, collaboration with contact tracing, quarantine of contacts of cases, and even a misbelief that measles virus circulation benefits the health of an individual. However, the limited number of clusters assessed is a limitation of this report (lines: 227-233).

Furthermore, we’ve added a sentence in the discussion:

In order to develop an effective strategy to improve confidence in vaccines in Austria, we need to understand the attitudes to vaccination in hesitant subgroups and conduct studies in the surrounding communities of clusters to examine their attitude towards vaccination, disease, and non-pharmaceutical interventions in general (lines: 244-247).

The mentioned notion in conclusions reads as follows:

This hypothesis on the vaccine attitude as one of the determinants for the magnitude of measles clusters has to be further tested (lines: 259-260).

Point 5: The authors have edited the manuscript to avoid the interpretation that these clusters were separate outbreaks but have not presented any new data to explain the size of the clusters.

Response 5: The size of the cluster A has been explained under the following lines: 141-147. The text reads as:

In Styria province, 35 identified cases belonged to cluster A with one transmission chain with four case generations. The primary case, with rash onset on 9 January 2019, was an unvaccinated school pupil of 15 years of age with an anthroposophical family background who consulted a paediatrician clinic in Graz, Styria during the contagious phase. In the same province, 34 successive cases occurred between 15 January and 20 February 2019 (Figure 1). Secondary (13 persons), tertiary and quaternary cases in the transmission chain occurred, but it was not possible to discriminate between the third and fourth generation cases.

And cluster B-F size under the lines 166-168. The text reads as:

In Salzburg province, between 10 January-01 March 2019, we identified 12 cases belonging to five further, unlinked clusters (clusters B-F) with chains of transmission which subsided after two generations (5 primary, 7 secondary cases).

Reviewer 3 Report

I think the article has been improved and the authors took into consideration the reviewer suggestions/recommendations. Therefore, the article can be published.

Author Response

Point 1: I think the article has been improved and the authors took into consideration the reviewer suggestions/recommendations. Therefore, the article can be published.

Response 1: Thank you for your kind feedback. To my understanding, there are no changes requested changes from your side.